# Robust Genetic Analysis of the X-Linked Anophthalmic (*Ie*) Mouse

**DOI:** 10.3390/genes13101797

**Published:** 2022-10-05

**Authors:** Brianda A. Hernandez-Moran, Andrew S. Papanastasiou, David Parry, Alison Meynert, Philippe Gautier, Graeme Grimes, Ian R. Adams, Violeta Trejo-Reveles, Hemant Bengani, Margaret Keighren, Ian J. Jackson, David J. Adams, David R. FitzPatrick, Joe Rainger

**Affiliations:** 1MRC Human Genetics Unit, Institute of Genetics and Cancer, University of Edinburgh, Crewe Rd South, Edinburgh EH4 2XU, UK; 2The Division of Functional Genetics and Development, The Roslin Institute, Midlothian EH25 9RG, UK; 3Wellcome Sanger Institute, Hinxton, Cambridgeshire CB10 1SA, UK

**Keywords:** anophthalmia, X-chromosome, X-ray induced allele, linkage analysis, *Zic3*, genome wide analysis, eye development

## Abstract

Anophthalmia (missing eye) describes a failure of early embryonic ocular development. Mutations in a relatively small set of genes account for 75% of bilateral anophthalmia cases, yet 25% of families currently are left without a molecular diagnosis. Here, we report our experimental work that aimed to uncover the developmental and genetic basis of the anophthalmia characterising the X-linked *Ie* (eye-ear reduction) X-ray-induced allele in mouse that was first identified in 1947. Histological analysis of the embryonic phenotype showed failure of normal eye development after the optic vesicle stage with particularly severe malformation of the ventral retina. Linkage analysis mapped this mutation to a ~6 Mb region on the X chromosome. Short- and long-read whole-genome sequencing (WGS) of affected and unaffected male littermates confirmed the *Ie* linkage but identified no plausible causative variants or structural rearrangements. These analyses did reduce the critical candidate interval and revealed evidence of multiple variants within the ancestral DNA, although none were found that altered coding sequences or that were unique to *Ie*. To investigate early embryonic events at a genetic level, we then generated mouse ES cells derived from male *Ie* embryos and wild type littermates. RNA-seq and accessible chromatin sequencing (ATAC-seq) data generated from cultured optic vesicle organoids did not reveal any large differences in gene expression or accessibility of putative cis-regulatory elements between *Ie* and wild type. However, an unbiased TF-footprinting analysis of accessible chromatin regions did provide evidence of a genome-wide reduction in binding of transcription factors associated with ventral eye development in *Ie*, and evidence of an increase in binding of the Zic-family of transcription factors, including Zic3, which is located within the *Ie*-refined critical interval. We conclude that the refined *Ie* critical region at chrX: 56,145,000–58,385,000 contains multiple genetic variants that may be linked to altered *cis* regulation but does not contain a convincing causative mutation. Changes in the binding of key transcription factors to chromatin causing altered gene expression during development, possibly through a subtle mis-regulation of Zic3, presents a plausible cause for the anophthalmia phenotype observed in *Ie*, but further work is required to determine the precise causative allele and its genetic mechanism.

## 1. Introduction

Anophthalmia affects 1 per 30,000 births [1] and is the severest structural eye malformation within the MAC phenotypic spectrum, that also includes Microphthalmia (small eye) and Coloboma. All are caused by disruptions to early eye organogenesis [2,3,4]. More than 90 genes have been associated with anophthalmia and microphthalmia (Reviewed in [2]), and the majority of causative variants disrupt transcription factors involved in early eye developmental events or pathways, such eye field specification and the retinoic acid signalling pathway [1,5,6]. Although some causative anophthalmia mutations follow a Mendelian inheritance pattern, a great proportion of non-syndromic anophthalmia associated mutations are sporadic de novo, with variable phenotypic expressivity even between immediate family members carrying the same mutation [1,2,7]. Despite the significant progress in the genetics research of severe ocular malformations, there is still a significant proportion of affected patients without an appropriate molecular diagnosis. Therefore, the investigation of new mechanisms or unexplored novel loci involved in the disruption of the normal process of eye development is important to extend the repertoire of anophthalmia genes [4,5] but more importantly, to expand our understanding of molecular mechanisms of eye diseases and their treatment or management.

The *Ie* mouse line is characterised by a combination of developmental eye and ear (*Ie)* malformations. The line was generated through an X-ray irradiation research programme in 1947 at Oak Ridge National Laboratory. The severe eye malformations segregated as an X-linked trait in a brief study published by Patricia Hunsicker [6], with details as follows. The disease locus was reported to be situated close to Bent tail (*Bn*) (since mapped to a loss of function mutation for *Zic3*) [7,8], and between Greasy (*Gs*)*,* and sparse fur (*Spf*) on chromosome X. Males and homozygous females displayed anophthalmia and malformations of the external ear. A broad phenotypic spectrum was observed in heterozygous females, from a barely noticeable eye phenotype to severe microphthalmia (small eye). Beyond the eye phenotype, the mice were viable and fertile [6]. We reasoned that this was an interesting anophthalmia allele, as additional alleles have never arisen spontaneously or through other mutagenesis screens, while the orthologous locus in humans has also not been associated with any structural eye malformations. We therefore set out to delineate the genetic basis of *Ie* to improve understanding for the genetic causes of anophthalmia.

## 2. Materials and Methods

### 2.1. Mice

The *Ie* mouse line was rederived and maintained in Biological Research Facilities at the University of Edinburgh. Animals were monitored regularly, with all husbandry and breeding performed according to the Animals (Scientific Procedures) Act 1986. All mouse work was undertaken according to study protocols approved within the Home Office project licences: 60/4424 and P1914806F (replacing 60/4424 in January 2018) under the University of Edinburgh establishment licence 60/2605 and subject to approval by institutional AWERB. Mouse eye processing for histology was performed as previously described [9].

### 2.2. High Molecular Weight DNA Isolation

Genomic high molecular weight DNA was isolated from 50 mg of kidney from *Ie* and wild type littermates. Tissue lysis was performed using 500 µL of 0.25% trypsin-EDTA (59428C, Sigma-Aldrich, MA, USA) and 5 mL of 1× red blood cell lysis buffer. Cell lysates were obtained using 3 mL lysis buffer (20 mM Tris-HCL pH7.5, 75 Mm NaCl, 50 mM EDTA pH8), 30 µL proteinase K PCR-grade (20 mg/mL) (03 115 879 001, Roche, Basel, Switzerland) and 150 µL Sarkosyl 20% (Sigma-Aldrich, MA, USA). A standard DNA isolation procedure using phenol:chloroform was performed. We used 1 µL DNAase-free RNase (Roche, Basel, Switzerland) to remove remaining RNA. DNA was purified using 1.2× Ampure XP beads. Genomic DNA was analysed for quality by electrophoresis with a 0.8% agarose gel stained with GelRed (ThermoFisher Scientific, MA, USA).

### 2.3. Whole Genome Sequencing

Genomic DNA samples were evaluated for quantity and quality using an AATI Fragment Analyzer and the DNF-487 Standard Sensitivity Genomic DNA Analysis Kit (Agilent, CA, USA). The AATI ProSize 2.0 software provided a quantification value and a quality (integrity) score for each sample. Based on quantification, gDNA samples were pre-normalised to fall within the acceptable range of the Illumina SeqLab TruSeq Nano library preparation method using Hamilton MicroLab STAR. Sequencing libraries were prepared using SeqLab specific TruSeq Nano High Throughput library preparation kits (Illumina, California, US) in conjunction with the Hamilton MicroLab STAR and Clarity LIMS X Edition. The samples were normalised to the concentration and volume required for the Illumina TruSeq Nano library preparation kits, then sheared to a 450 bp mean insert size using a Covaris LE220 focused-ultrasonicator. The inserts were ligated with blunt ended, A- tailed, size selected, TruSeq adapters and enriched using 8 cycles of PCR amplification. The libraries were evaluated for mean peak size and quantity using the Caliper GX Touch with a HT DNA 1k/12K/HI SENS LabChip and HT DNA HI SENS Reagent Kit and normalised to 5nM using the GX data and the actual concentration established using a Roche LightCycler 480 (Roche, Basel, Switzerland) and a Kapa Library Quantification kit and Standards (Roche, Basel, Switzerland).

The libraries were normalised, denatured, and pooled in eights for clustering and sequencing using a Hamilton MicroLab STAR with Genologics Clarity LIMS X Edition. Libraries were clustered onto HiSeqX Flow cell v2.5 on cBot2s and the clustered flow cell is transferred to a HiSeqX for sequencing using a HiSeqX Ten Reagent kit v2.5.

Demultiplexing was performed using bcl2fastq 2.17.1.14, allowing 1 mismatch when assigning reads to barcodes. Samples were quality checked with FastQC (https://www.bioinformatics.babraham.ac.uk/projects/fastqc/ (accessed on 4 August 2022)), aligned to the GRCm38 assembly with bwa 0.7.13-r1126 [10] and duplicates marked with biobambam2 2.0.44 (https://github.com/gt1/biobambam2 (accessed on 4 August 2022)). Indel realignment, base quality score recalibration, and GVCF generation was performed with the Genome Analysis ToolKit (GATK) 3.4.0 [11]. Joint genotyping of samples was performed with GATK 4.0.2.1 and annotated with the Ensembl Variant Effect Predictor v94 [12] and the Mouse Genome Project version 6 database [13].

### 2.4. Long-Read Sequencing

High molecular weight DNA quality was analysed using the Agilent 2200 TapeStation system (Agilent, California, US) and by pulse-gel electrophoresis. Subsequently, samples were sequenced using Oxford Nanopore Technologies at Edinburgh Genomics facility. Long-read sequencing analysis was performed using nano-snakemake [14] where an average read length of ~14 Kb was obtained. Subsequently, we performed structural variant (SV) analysis using Sniffles (https://github.com/fritzsedlazeck/Sniffles (accessed on 4 August 2022)) with the same parameters as the Decode nanopore sequencing report [15]. Then, a joint called using SURVIVOR (https://github.com/fritzsedlazeck/SURVIVOR (accessed on 4 August 2022)) [16].

### 2.5. Derivation of mESCs

Mouse embryonic stem cells were derived from the *Ie* mouse and maintained in 2i media as previously described [17]. E2.5 embryos were flushed from oviducts in M2 media (Sigma) and maintained overnight in M16 media (Sigma-Aldrich®, MA, USA) supplemented with 1 µM PD0325901, 3 µM CHIR99021, and 1× penicillin/streptomycin. Subsequently, blastocysts were maintained in 2i media for two days followed by isolation of inner cell mass by immunosurgery. The inner cell mass was maintained in 2i media for 5–7 days until cell outgrowths were dissociated and expanded following standard cell culture procedures.

### 2.6. DNA Isolation from mESCs

Cell lysates were obtained adding 50 µL of lysis buffer (see above) to mESCs cultures in 32 mm^2^ wells, followed by incubation for 14 h at 55 °C and 1.5 h at 85 °C. We added 1 µL of cell lysate directly to PCR.

### 2.7. Sex Determination and Genotyping of mESCs

Mouse embryonic stem cells derived from the *Ie* mouse were genotyped with the following oligonucleotides: *Vgll1* Forward: 5′-CCTGAAAATGGTGCCAGAAG-3′, *Vgll1* Reverse 5′-CATGAGCGATCCTGTGCTT-3′, *Ie*_mut Forward 5′-GCTATACACACAGATGGATCCA-3′ and *Ie*_mut Reverse 5′-GCTTCTGAATTATAATCTTTCAT-3′. Previously published PCR oligonucleotides specific for *Rbm31x* and *Rbm31y* were used for sex determination [18].

### 2.8. Optic Organoids Culture

Optic vesicle 3D culture was performed following the procedure described previously [19]. The Rax-GFP cell line is a derivative from E14Tg2A from the 129/Ola mouse strain. Bulk RNA was isolated using the RNeasy^®®^ mini columns (QIAGEN, Düsseldorf, Germany) following the manufacturer’s instructions. Subsequently, cDNA was synthesized from 150 ng of total RNA with AffinityScript RT buffer (Agilent, CA, USA) according to the manufacturer’s instructions. TaqMan assays used the following probes, *Rax:* Mm01258704_m1; *Pax6:* Mm00443081_m1 for qPCR analysis using the Roche LightCycler 480 System.

### 2.9. Statistical Analysis

Analyses of data derived from qPCR were performed using ‘PCR’ package in Rstudio [20]. Non-parametric Wilcoxon test for independent samples was performed, where a normal distribution and homogeneous variance was not assumed.

### 2.10. RNA-seq

Libraries for RNA-seq analysis were prepared using 500 ng of total RNA from three independent replicates using NEBNext Ultra II Directional RNA library prep kit for Illumina (NEB) and PolyA mRNA magnetic isolation module (NEB). Sequencing was performed using NextSeq 500/550 High-Output v2.5 (150 cycle) kit on NextSeq 550 platform (Illumina, San Diego, US). Libraries were combined into a single equimolar pool of 12, based on Qubit and Bioanalyser results and run across a High Output v2.5 Flow Cell.

RNA was sequenced in the Clinical Research Facility in the Welcome Trust Unit at the Western General Hospital of Edinburgh. Data quality control was performed using FastQC: http://www.bioinformatics.babraham.ac.uk/projects/fastqc (accessed on 4 August 2022) and cutadapt: http://cutadapt.readthedocs.io/en/stable/ (accessed on 4 August 2022). Reference genome indexing was done using STAR: https://github.com/alexdobin/STAR (accessed on 4 August 2022), and read mapping with SAMtools: http://www.htslib.org/ (accessed on 4 August 2022). Subsequently, featureCounts: http://subread.sourceforge.net/ (accessed on 4 August 2022) was used for gene expression quantification. To perform differential expression analysis edgeR, pheatmap and PCAtools packages were used in RStudio v1.3.1093 (Rsutdio, PBC. Boston, USA) [21,22].

### 2.11. ATAC-seq

ATAC-seq samples and libraries were generated from two independent biological replicates of optic vesicle cultures for each genotype as previously described [23]. Library quality was assessed using Agilent Bioanalyser (Agilent Technologies, #G2939AA) with DNA HS kit (#5067-4626). Sequencing was performed using the NextSeq 500/550 High-Output v2.5 (150 cycles) kit (#20024907) on the NextSeq 500/550 platform (Illumina). Two pools of four libraries were sequenced on high-output flow cells.

### 2.12. ATAC-seq Mapping and Analysis

To perform initial quality control, pre-processing, alignment, peak-calling and quantification we used the nf-core ATAC-seq pipeline [24] with default parameters. The full pipeline details can be found at https://nf-co.re/atacseq (accessed on 4 August 2022), but we highlight that sequencing reads were aligned to the mm10 mouse genome and in this work we used consensus broad peaks across replicates called using MACS2 [25]. Differential accessibility of peak-regions was performed using DEseq2 [26]. To quantify putative differences in TF binding we used the TOBIAS framework [27]. As input to TOBIAS we used the BAM files from merged wild type and mutant replicates, and we used the *Mus Musculus* motifs from the JASPAR2020 database [28] to identify putative binding sites and compute footprint scores within the consensus peak set. In detail, we use TOBIAS-ATACorrect to correct for Tn5-insertion bias, followed by TOBIAS-ScoreBigwig to compute base-pair footprint scores across the selected input peak regions. We then use the latter results together with the JASPAR2020 motif database, to identify occurrences of motifs, compute associated footprint scores and quantify differences in putative binding of TFs, using the TOBIAS-BINDetect. Finally, we use TOBIAS-PlotAggregate to generate aggregate footprint results—corrected ATAC-seq signal averaged across all detected occurrences—for any input TF-motif. To visualise ATAC-seq results (differential binding and footprints) we have used the Matplotlib Python library [29].

## 3. Results

### 3.1. Phenotyping Ie Ocular Defects

We rederived the *Ie* mouse onto a C57BL/6J inbred strain background (Appendix A) and observed the anophthalmia and external-ear phenotypes in adults as previously described [9] (Figure 1A). We then examined *Ie* eyes at embryonic stages and observed that the phenotype was characterized by severe structural defects at E13.5, including hypoplastic retina and lens (Figure 1B–E, Appendix A). Examination of eyes from earlier embryonic stages indicated that development of the optic vesicle in *Ie* was apparently normal at E9.5, but the optic cup had become defective by E11.5, when ventral retinal structures were underdeveloped and the dorsal RPE was abnormally thickened (Appendix A). Surprisingly, lens development appeared comparable between wild type and *Ie* at E11.5 (Appendix A), suggesting that degeneration of early retinal tissues may underlie the subsequent broader eye phenotype.

### 3.2. Genetic Mapping Ie

*Ie* was identified from the progeny of an irradiated male mouse—an F1 from a 101 (female) x C3H (male) cross (Appendix A). Female *Ie*^/+^ offspring were then crossed to C3H for multiple generations (P Hunsicker, personal communication). Previous linkage-based mapping positioned the *Ie* locus to the X chromosome, between the Gs (Greasy) and Spf (Sparse fur, a mutation in Otc, Ornithine transcarbamylase) loci, close to Bn (Bent tail). Bent tail has been identified as a ~60 to 170 kb deletion on the X-chromosome in a gene desert that includes the Zic3 gene [10,11]. It was therefore concluded that *Ie* had arisen on 101 near *Zic3*, but was not phenotypically consistent with a loss-of-function mutation for *Zic3* [9]. After rederivation, phenotypically heterozygous females (i.e., visible microphthalmia) were serially backcrossed with wild type C57BL/6J males to enable meiotic recombination and therefore reduce the background 101 region to facilitate genetic mapping.

We began our mapping strategy using a panel of microsatellite and SNP markers on chromosome X that extended ~5 Mb on either side of the *Zic3* locus. We found three speculative candidate regions between single informative SNP markers (Figure 2A). These were, region 1: rs33880149–rs13483760 (GRCm39, chrX:48,737,129–54,971,490); region 2: rs13483761–rs13483770 (chrX:55,261,254–58,280,858); and region 3: from rs29058690–rs29051707 (chrX:58,352,618–66,158,608). At this point we were prevented from continuation of this strategy due to a lack of informative and strain-specific microsatellite and SNP markers available in published online genomic databases, and difficulties in reliably sourcing 101 genomic DNA. Instead, we adopted short-read whole genome sequencing (WGS) approach using genomic DNA from phenotypic *Ie* mutant (X(*Ie*)/y) and wild type (X(wt)/y) male littermates.

### 3.3. Whole Genome Sequencing

First, we used short read WGS data to refine the candidate interval. We used DNA obtained from an affected (*Ie)* and unaffected (wild type) males and generated paired-end sequencing libraries of 150 bp average read length. This analysis allowed us to identify variants in *Ie* that were absent from wild type and not seen in other mouse strain genotypes available from the Mouse Genome Project. Using these, we mapped the *Ie* critical interval to a 6 Mb region at chrX:56,145,000–58,385,000 (Figure 2B). Within this large region, the WGS data identified two separate internal regions with high levels of sequence variation between *Ie* and Wt: (i) a centromeric variant cluster chrX:56,795,360–57,255,360 and (ii) a telomeric variant cluster chrX:57,965,360–58,075,360. Within these regions were the coding genes *Zic3* and *Fgf13*, respectively (Figure 2C).

We then used the same data to identify potential causative non-synonymous changes in coding exons for all genes within the larger *Ie* critical interval. Only one intragenic coding mutation was identified, a 39 bp in-frame deletion in exon 6 of *Vgll1* (c.606_644 del39; p. Asp203_Pro215del) that was present in *Ie* but not in Wt mouse DNA. *Vgll1* encodes for a transcriptional coactivator that interacts with TEAD proteins [30,31]. In humans, *VGLL1* is expressed in different embryonic tissues such as the lung, heart, and placenta [32], but no evidence of activity in the eye has so far been reported. However, in further analysis we found this same deletion was present in genomic DNA sequences obtained from wild type E14 mouse ES cells from the inbred mouse strain 129/Ola. Thus, this lesion was classified as benign. No other variants were identified within coding exons, UTRs, or splice-sites for the coding genes contained within the large candidate interval, including *Zic3* and *Fgf13*.

### 3.4. Long-Read Genomic DNA Sequencing

As we were unable to find any coding mutations in *Ie*, we then reasoned that the causative mutation may be a genomic rearrangement event, especially as exposure to X-ray irradiation is often the cause of structural genetic rearrangements [33], and our previous work found a balanced chromosomal inversion as the cause of a mouse line with a severe eye malformation that had arisen as part of a similar radiation induced strategy [9]. Therefore, to further investigate the genetic basis of anophthalmia in *Ie*, we generated long-read sequencing (Oxford Nanopore Technologies) to enable both coding mutation analysis (small indels, substitutions) and the identification of structural variants (SVs; inversions and large deletions or insertions). From these analyses, 22 SVs were identified that were specific to *Ie* within the chrX large critical interval (Appendix A). However, these SVs were either intergenic or intronic mutations, five of which had been already reported in the mouse genome project (https://www.sanger.ac.uk/data/mouse-genomes-project/ (accessed on 4 August 2022)).

### 3.5. Expression Analyses

As no clearly causative genetic lesion that segregated exclusively with *Ie* had been identified through mapping and whole genome sequencing strategies, we chose to examine differences at the gene expression level between wild type and *Ie*, testing the hypothesis that changes to gene expression levels may implicate a genetic pathway or mechanism for *Ie*. Mouse embryonic stem cells (mESCs) were derived from a cross between an affected *Ie* female and a wild type C57BL6/J male (Figure 3A). Wild type and mutant male mESCs were then used to generate optic vesicle (OV) organoids (Figure 3A), following the method described previously [19]. Mutant and Wt organoids were imaged at day 7 and compared to additional control organoids derived from Rax-GFP mESCs, which allowed us to monitor and confirm OV development using the retinal-specific *Rax* driven GFP expression as a reporter for successful OV development (Figure 3B). *Pax6* and *Rax* expression were also measured in OV organoids by qRT-PCR (Figure 3C) and confirmed the expected retinal differentiation compared to uninduced mESCs. No significant differences of expression for these two marker genes were observed between Wt and *Ie*. To then identify gene expression changes in the critical interval, RNA-seq analysis was performed at day five of OV culture in pools of forty-five organoids per sample and using two independent biological replicates (independently derived cell lines) for each genotype. Principal component analysis revealed high variability between the two mutant biological replicates (Appendix A), with one sample showing the presence of detectable alleles indicative of contaminating non-source mESCs. Therefore, subsequent analyses were performed using technical replicates from OV culture from one mutant and two wild type lines (Figure 3D). No significant differences in gene expression were observed for eye field transcription factors, pluripotency, or neural markers between the genotypes (FDR > 0.01 obtained by multiple test correction. Genes located within the critical interval identified in our earlier microsatellite and SNP analyses were then assessed (Figure 3D) for differential expression and indicated potential decreased expression in *Ie* for *Zic3*, *Map7d3*, and *Mospd1* but these gene expression changes failed to reach significance. We then asked whether existing transcriptomic data could support the identification of candidate genes in this broader *Ie* critical interval; mRNA-seq data of laser-capture dissected tissue from dorsal and ventral eye during optic fissure closure stages (E11.5–E12.5) from Patel et al. [34] was used to determine expression levels (Appendix A). Those genes with expression levels determined to be physiologically relevant in either dorsal or ventral eye were: *Fgf13*, *Fhl1*, *Htatsf1*, *Ints6l*, *Mmgt1*, *Rbmx*, *Zfp449*, and *Zic3*. Thus, although no mutations or significant gene expression changes were identified in *Ie* for these genes, they could be considered as potential candidates for further analyses arising from this work.

### 3.6. ATAC-seq

Lastly, we wanted to investigate the differences between affected and unaffected *Ie* mouse at chromatin accessibility level. To do so, an ATAC-seq analysis was performed in parallel to the RNA-seq analysis, at day five of optic vesicle organoid culture (Figure 4A). Focusing on the critical regions within chromosome X, we found no evidence of differentially accessible peaks (DeSeq2, adjusted-*p* value < 0.01). Extending our search to look for differences in regulatory element accessibility genome-wide, we found only a handful (12×) of such peaks, all located on chromosome 14. The two most significant of these peaks (both relatively depleted in the mutant samples) overlap the promoters of *Uchl3* and *Pibf1*, which have been associated with retinal layer [35], and lens morphology defects according to the mouse phenotyping consortium [36]. We noted that no differentially accessible peaks are found within a window of 1 Mb around each of the canonical eye-field genes. This could be evidence towards regulation of eye-field initiation either being unaffected or only very slightly affected in the *Ie* mouse, as far as chromatin accessibility of their driver regulatory elements is concerned. Because we found no strong distinguishing chromatin peak signals between wild type and mutant, we investigated whether our ATAC-seq datasets could provide evidence of more subtle effects and in particular differences in TF binding. To do this we used genomic-footprinting analyses, which exploit the fact that the presence of bound TFs can protect DNA against transposase cleavage, resulting in relative decreases in accessibility signals within accessible regulatory elements.

Using the TOBIAS framework [27], we computed base-pair footprint scores (relative depletion inaccessible chromatin) for consensus peaks in the wild type and mutant.. Differences between these footprint scores at TF-motif occurrences within consensus peaks were then used to compute differential binding scores for each mouse motif available in the Jaspar database [28]. Interestingly, amongst the motifs displaying higher binding scores in *Ie* mutant compared to wild type, are a group of motifs belonging to the Zic-family of TFs (Figure 4B), including Zic3 which lies within the identified X-chromosome critical region. The same analysis suggests a relative depletion of binding of TFs important in ventral structure development, such as Vax1, Vax2 and Hes1, as well as Sox2 (known causative gene for eye-malformations) in the mutant (Figure 4B). This may be evidence that these TFs do not bind their cognate motifs as strongly at regulatory elements controlling ventral-TF target gene expression and aligns with the ventral morphological defects observed in the *Ie* mouse. These apparent differences in binding between *Ie* and wild type can be made visually discernible by looking at the aggregate ATAC-seq signal (corrected for Tn5 bias) around the TF-motif occurrences and in particular the difference between depletion in signal around the motif centre and in the regions flanking the motif. Indeed, the aggregate signal around Zic3 motif occurrences (Figure 4C) indicate a noticeably deeper footprint (evidence of stronger binding) in the mutant compared to the wild type samples, and shallower footprints (evidence of weaker binding) for TFs such as Sox2 and Vax1 (Appendix A).

This was made further apparent by looking at the aggregate ATAC-seq signal (corrected for Tn5 bias) around the Zic3 motifs, which illustrates a deeper binding footprint in the mutant compared to the wild type (Figure 4C). Additionally, the differential footprinting analysis suggests a depletion of binding of TFs important in ventral structure development, such as Vax1, Vax2 and Hes1, as well as Sox2 (known causative genes for eye-malformations) in the mutant (Figure 4A, Appendix A). This may be evidence that these TFs do not bind their cognate motifs as strongly at regulatory elements controlling ventral-TF target gene expression and aligns with the ventral morphological defects observed in the *Ie* mouse.

## 4. Discussion

Genetic factors are recognised as the major cause of developmental eye malformations. Even though a significant number of eye disease-causing mutations have been reported, a considerable proportion of patients with eye structural malformations remain without a molecular diagnosis [4,37]. In this study, we investigated the basis of the X-ray induced anophthalmia in the *Ie* mouse as a potential novel locus and to identify additional genetic mechanisms involved in the disruption of eye development.

Our first analyses suggested that the causal mutation is non-coding and potentially a chromosomal rearrangement derived from the radiation exposure. However, subsequent analyses in short and long read WGS did not show any plausible coding mutations or structural rearrangements that may indicate causality on the eye phenotype. It has been reported previously how exposure to ionizing radiation can result in a significant increase in de novo single-nucleotide variants and indels with a notable over-representation of clustered mutations [38,39]. From our analyses, we found an imbalance of mutations across the *Ie* mouse genome, particularly within the critical region in chrX (Appendix A). We hypothesise that the multiple de novo single-nucleotide mutations found within the centromeric and telomeric variant clusters in the critical region of chrX may be involved in the severe ocular phenotype.

Previous studies have shown that the transcription factor Zic3 is involved in developmental decisions during early embryogenesis, including patterning of the anterior visceral endoderm, gastrulation induction, and positioning of the primitive streak. In humans, *ZIC3* mutations are associated with X-linked heterotaxy (MIM 306955) [40]. Multiple Zic binding sequences have been identified [41], and we found evidence for an enrichment of Zic3 motif binding in ATAC-seq data during optic vesicle differentiation from ESCs. However, whether Zic TFs act as transcriptional activators or repressors is poorly understood [42]. Our data suggest that there is an increase in binding affinity of the Zic-family in the mutant compared to the wild type, yet how this is associated with eye structural malformations remains unclear. The *Ie* locus was originally mapped using classical mouse genetics strategies, [9], with crosses to *Greasy* (Gs), *Blotchy* (Blo), *Bent tail* (Bn) and *sparse fur* (Spf), all of which are located on chromosome X. Remarkably, in over 2,000 males obtained from crosses only one recombination event was observed between *Bn* and *Ie*, but this male was unfortunately not kept for further mapping (P. Hunsiker, personal communication). The *Bn* locus has since been refined to a <170 kb deletion that includes the entire coding region for *Zic3* [11] providing strong evidence that the *Ie* locus is close to *Zic3* on chrX and consistent with our data presented here. Our analysis of existing transcriptomic data found that *Zic3* is expressed in the developing eye, however given the disparity between the phenotypes of these two lines, *Ie* is unlikely to be a loss of function allele for this gene. As our in silico data suggest increased binding of Zic TFs at a genome-wide level, one plausible mechanism for the *Ie* phenotype is that the ancestral DNA contains an allele that causes increased expression of *Zic3*. Increased levels of Zic3 protein and transcription factor activity could affect the expression of genes whose dosage is critical during development of the ventral optic cup. More refined transcriptional analyses of ocular tissues from *Ie* are required to confirm this hypothesis.

As a follow up to the evidence provided by our ATAC-seq analyses it could be very useful to perform Cut and Run experiments on the organoids targeting DNA-bound transcription-factors highlighted from our ATAC-seq data. This would allow us to validate the hypotheses of stronger/weaker binding of Zic3 and Sox2 in the *Ie* mutant compared to wild-type, as well as investigating whether there is redistribution of these important TFs recruited at different regulatory elements. This would also potentially uncover currently unknown genes involved in ventral retina development. To provide more insight into the consequence rather than the cause of the *Ie* genetic lesion at the gene expression level, it would also be informative to conduct transcriptomic analyses of resected ventral retina, and to include analyses at the single-cell level. Indeed, scRNAseq could also be applied to the organoid cultures to overcome any heterogeneity among the cells comprising these samples.

We conclude that despite the application of a range of mapping and sequencing approaches, and the use of gene expression and regulation analyses, we have been unable to reveal the causative mutation for the developmental eye and ear defects observed in the *Ie* mouse. Nevertheless, our data provides a useful framework for others to build upon to identify the genetic cause of *Ie*, and to reveal the precise molecular mechanism of the severe developmental anophthalmia found in this unique and enigmatic mouse line.

## Figures and Tables

**Figure 1 genes-13-01797-f001:**
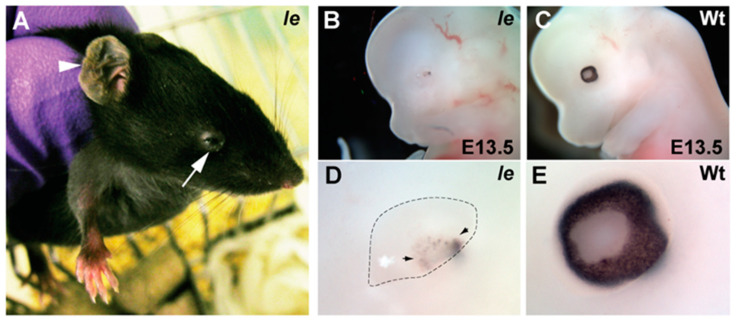
*Ie* gross phenotyping. (**A**) Adult *Ie* mice display abnormal pinnae (arrowhead) and small eyes (arrow). (**B**) Whole embryo analysis at E13.5 revealed severe ocular malformations in mutant eyes (**B**) and enlarged in (**D**), compared to stage-matched wild types (**C**) and enlarged in (**E**). Arrowheads indicate pigmented cells within the hypoplastic eye.

**Figure 2 genes-13-01797-f002:**
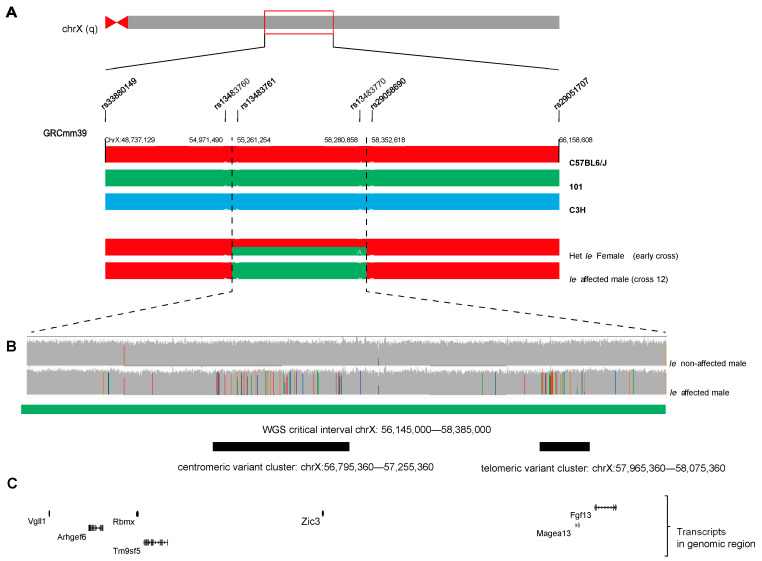
Interval mapping. (**A**) Schematic of chromosome X. Highlighted region mapped using a panel of microsatellite markers and SNPs (rs33880149 [chrX:48,737,129] to rs29051707 [chrX:66,158,608]; GRCm39). Parental chromosomes C57BL/6J (red), 101 (green) and C3H (blue) are shown. Chromosome regions of a heterozygous female (early cross) and an affected male (cross 12) displaying three mapped *Ie* intervals within this region. (**B**) Short-read whole genome sequencing (WGS) alignment for male wild type (non-affected) and *Ie* mutant (affected) littermates. Critical interval: chrX: 56,145,000–58,385,000 (green) and multi-nucleotide centromeric and telomeric variant clusters (black) present in mutant mouse. (**C**) Annotated transcripts within critical interval.

**Figure 3 genes-13-01797-f003:**
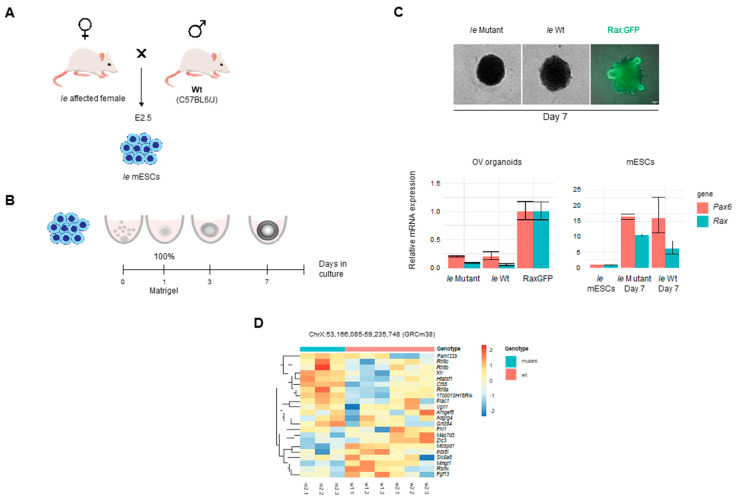
*Ie* mouse embryonic stem cells derivation and gene expression analyses. (**A**) Schematic of mouse breeding between a heterozygous affected *Ie* female and a wild type C57BL/6J male. ES cells were isolated from the inner cell mass of E2.5 blastocysts. (**B**) OV culture from day 0 to day 7. (**C**) OV at day 7, *Ie* mutant and *Ie* wild type organoids shown in bright field, and Rax:GFP-derived organoids used as a positive control. Forty-five organoids per group were used in triplicate to measure gene expression by qPCR. Bar plot on the left shows Rax and Pax6 expression in *Ie* mutant and *Ie* wild type OV at day 7 relative to Rax:GFP positive controls. Bar plot on the right shows Rax and Pax6 expression in mutant and wild type organoids at day 7 relative to undifferentiated mESCs. (**D**) Heat map of RNA-seq data depicting differential expression of genes within the initial critical interval in chrX between triplicates of mutant (m^2^) and wild type (w1 and w2) OV culture samples.

**Figure 4 genes-13-01797-f004:**
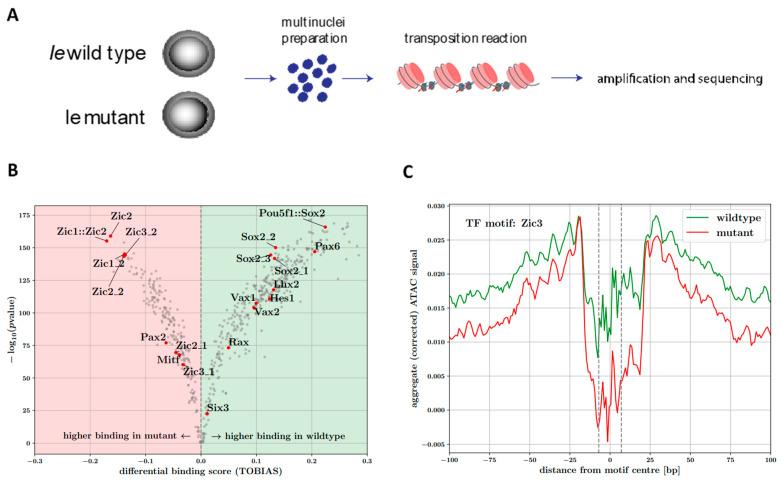
ATAC-seq analysis in optic vesicle organoids. (**A**) Diagram summarizing chromatin accessibility analysis from OV organoids. (**B**) Volcano plot of TOBIAS differential binding analysis. Positive and negative scores are evidence of increased TF binding across consensus sets of peak regions, in wild type and mutant samples, respectively. Red points highlight results for motifs of TFs known to be important in eye-field specification and ventral ocular structure development. (**C**) Zic3 motif footprint in wild type and mutant samples. Aggregate Tn5-bias corrected ATAC-seq signal around detected Zic3 motif occurrences in consensus peak regions for wild type (green line) and mutant (red line) samples. Vertical dashed lines indicate edges of the Zic3 motif.

## Data Availability

All sequence data (ATAC, RNAseq, WGS) for this study have been deposited in the European Nucleotide Archive (ENA) at EMBL-EBI under accession number PRJEB55172.

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
