# Peer review of "Robust Genetic Analysis of the X-Linked Anophthalmic (Ie) Mouse"

_genes, 2022, doi:10.3390/genes13101797_

Round 1
Reviewer 1 Report
the manuscript is well written and the study is well conducted, however, the discussion lacks of the study limitations and the conclusion should be improved, maybe adding some hints for future studies.
Author Response
Reviewer 1
The manuscript is well written, and the study is well conducted, however, the discussion lacks of the study limitations and the conclusion should be improved, maybe adding some hints for future studies.
We appreciate the encouragement from the reviewer and the invitation to add more information, in particular towards potential next steps for this interesting mouse line. We would have added this statement to the end of the discussion:
As a follow up to the evidence provided by our ATAC-seq analyses it could be very useful to perform Cut & Run experiments on the organoids targeting DNA-bound transcription-factors highlighted from our ATAC-seq data. This would allow us to validate the hypotheses of stronger/weaker binding of Zic3 and Sox2 in the Ie mutant compared to wild-type, as well as investigating whether there is redistribution of these important TFs recruited at different regulatory elements. This would also potentially uncover currently unknown genes involved in ventral retina development. To provide more insight into the consequence rather than the cause of the Ie genetic lesion at the gene expression level, it would also be informative to conduct transcriptomic analyses of resected ventral retina, and to include analyses at the single-cell level. Indeed, scRNAseq could also be applied to the organoid cultures to overcome any heterogeneity among the cells comprising these samples.
Reviewer 2 Report
Review genes-1878425
In this study, the authors investigated an unexplored novel locus known as Ie. le is an X-linked genetic disorder known to cause abnormalities in the ear and eye development, but the mechanism by which it causes the disease is still unknown. A series of studies (locus mapping, sequencing, gene expression levels) conducted by the authors could not show problems with cis-acting elements as well as mutations in specific genes present in the le locus. Only the TF binding footprint results based on bioinformatics showed that the binding of the Zic-family to the mutant Ie locus is increased. Although the authors could not clearly identify the causative factors present in le locus, the direction of the study and the research method performed seem to be generally appropriate. However, it was difficult to understand the text due to the lack of detailed explanations for the cited study. Also, some figures seem crude enough to not match the quality of the publication. Therefore, it is suggested to correct the mentioned issues.
Comment 1:
Line # 50 - 51 in Introduction. “The majority of mutation ~ de novo with variable expressivity” : The description of the cited study is too concise, making it difficult to understand what the author is explaining.
Comment 2
Line # 238 – 240 in Result, “The original irradiated male ~ a loss of function mutation for Zic3” : It is very hard to understand the explanation of cited results. This sentence is thought to be unnecessary if it is not related to the context.
Comment 3
Line # 244 – 257 in Result and Figure 2. “We began our mapping ~ wild type littermate” : it is thought that the picture is crude and does not reflect the description of the text well.
Comment 4
Figure 4C and supplement figure S3 : Both the Zic3 motif footprint and the Sox2 motif footprint show that the graphs are very similar to each other. If so, does the mutant le not only increase the Zic3 binding but also the Sox2 binding? I think the analysis of this part is insufficient.
Author Response
Reviewer 2
In this study, the authors investigated an unexplored novel locus known as Ie. le is an X-linked genetic disorder known to cause abnormalities in the ear and eye development, but the mechanism by which it causes the disease is still unknown. A series of studies (locus mapping, sequencing, gene expression levels) conducted by the authors could not show problems with cis-acting elements as well as mutations in specific genes present in the le locus. Only the TF binding footprint results based on bioinformatics showed that the binding of the Zic-family to the mutant Ie locus is increased. Although the authors could not clearly identify the causative factors present in le locus, the direction of the study and the research method performed seem to be generally appropriate. However, it was difficult to understand the text due to the lack of detailed explanations for the cited study. Also, some figures seem crude enough to not match the quality of the publication. Therefore, it is suggested to correct the mentioned issues.
Comment 1
Line # 50 - 51 in Introduction. “The majority of mutation ~ de novo with variable expressivity” : The description of the cited study is too concise, making it difficult to understand what the author is explaining.
We apologise for the lack of clarity in this sentence, which prevented its correct interpretation. We have amended this to improve clarity.
** Original text **
Although some causative anophthalmia mutations follow a Mendelian inheritance pattern, the majority of mutations associated with non-syndromic anophthalmia are sporadic de novo with variable expressivity [1,2,7].
** Updated text **
Although some causative anophthalmia mutations follow a Mendelian inheritance pattern, a great proportion of non-syndromic anophthalmia associated mutations are sporadic de novo, with variable phenotypic expressivity even between immediate family members carrying the same mutation [1,2,7].
Comment 2
Line # 238 – 240 in Result, “The original irradiated male ~ a loss of function mutation for Zic3” : It is very hard to understand the explanation of cited results. This sentence is thought to be unnecessary if it is not related to the context.
The purpose of this sentence was to draw the readers attention to two key points: (1) that the Ie allele arose on DNA from the 101 mouse strain background, and (2) that the mutation is not consistent with a loss of function allele for Zic3 based on the published Zic3 knockout mouse data available. However, we agree that this sentence as written was superfluous as this information is detailed in the previous part of the paragraph. We have updated the text accordingly.
** Original text **
“The original irradiated male must have carried the X-chromosome from strain 101 and it was concluded that the Ie allele had arisen on 101, near Zic3, but was not a loss-of-function mutation for Zic3 [9]”
** Updated text **
“It was therefore concluded that Ie had arisen on 101 near Zic3, but was not phenotypically consistent with a loss-of-function mutation for Zic3 [9]”
Comment 3
Line # 244 – 257 in Result and Figure 2. “We began our mapping ~ wild type littermate” : it is thought that the picture is crude and does not reflect the description of the text well.
We apologise for the oversights with this figure and that the main text did not convey the data presented, specifically that for clarity we only showed critical interval 1 identified through the initial SNP/microsatellite mapping. We also noted a mistake regarding the naming of the SNPs in Fig2A, specifically double annotation of rs13483760. We have amended the figure and the legend to improve both these issues.
** Original text **
We began our mapping strategy using a panel of microsatellite markers and SNPs on chromosome X (rs33880149 [chrX:48,737,129] to rs29051707 [chrX:66,158,608]; GRCm39) (Figure 2A) that extended ~5 Mb on either side of the Zic3 locus (chrX:57,076,003-57,081,919). Although most of these markers were uninformative between C57BL/6J control DNA samples and Ie, we found three speculative candidate regions (Figure 2A) between single informative SNP markers that were unambiguously C57B/L6 in the Ie DNA samples. These were, region 1: rs30428151-rs13483760 (GRCm39, chrX:52,255,212-54,971,490); region 2: rs13483760-rs13483770 (chrX:54,971,490-58,280,858); and region 3: from rs13483770 extending in the telomeric region beyond rs29053241 (chrX:58,280,858-61,690,566). At this point we were prevented from continuation of this strategy due to a lack of unambiguous and discriminatory strain-specific markers available in published online genomic databases, and difficulties in reliably sourcing 101 genomic DNA. Instead, we adopted short-read whole genome sequencing (WGS) approach using genomic DNA from phenotypic Ie mutant (X(Ie)/y) and wild type (X(wt)/y) male littermates.
** Revised text **
We began our mapping strategy using a panel of microsatellite and SNP markers on chromosome X that extended ~5 Mb on either side of the Zic3 locus. We found three speculative candidate regions between single informative SNP markers (Figure 2A). These were, region 1: rs33880149-rs13483760 (GRCm39, chrX:48,737,129-54,971,490); region 2: rs13483761-rs13483770 (chrX:55,261,254-58,280,858); and region 3: from rs29058690--rs29051707 (chrX:58,352,618-66,158,608). At this point we were prevented from continuation of this strategy due to a lack of informative and strain-specific microsatellite and SNP markers available in published online genomic databases, and difficulties in reliably sourcing 101 genomic DNA. Instead, we adopted short-read whole genome sequencing (WGS) approach using genomic DNA from phenotypic Ie mutant (X(Ie)/y) and wild type (X(wt)/y) male littermates.
We also updated Figure 2, improving resolution and repairing an error, and also revised the figure legend to better reflect the data displayed.
Figure 2. Interval mapping. (A) Schematic of chromosome X. Highlighted region mapped using a panel of microsatellite markers and SNPs (rs33880149 [chrX:48,737,129] to rs29051707 [chrX:66,158,608]; GRCm39). Parental chromosomes C57BL/6J (red), 101 (green) and C3H (blue) are shown. Chromosome regions of a heterozygous female (early cross) and an affected male (cross 12) displaying three mapped Ie intervals within this region. (B) Short-read whole genome sequencing (WGS) alignment for male wild type (non-affected) and Ie mutant (affected) littermates. Critical interval: chrX: 56,145,000-58,385,000 (green) and multi-nucleotide centromeric and telomeric variant clusters (black) present in mutant mouse. (C) Annotated transcripts within critical interval.
Comment 4
Figure 4C and supplement figure S3 : Both the Zic3 motif footprint and the Sox2 motif footprint show that the graphs are very similar to each other. If so, does the mutant le not only increase the Zic3 binding but also the Sox2 binding? I think the analysis of this part is insufficient.
We thank the referee for their comment. This made it apparent to us that these paragraphs require further clarification to aid comprehension and have therefore re-written parts of the associated text to address this. We note that the aggregate footprint plots for Zic3 (Figure 4C) and Sox2 (Supplementary Figure 3) are consistent with the differential binding analysis of Figure 4B, namely that there is evidence of stronger binding of Zic3 and weaker binding of Sox2 in the Ie mutant compared to wildtype. The key aspect of the aggregate ATAC-seq plots is the depth of the motif footprint, namely the difference between the signal around the motif centre, and the signal in the regions flanking the motifs. We have updated the text to make clarify this for the readers. We have additionally included a further aggregate plot (for Vax1) to the supplementary figure 3 for completeness.
** Original paragraph text **
Because we found no strong distinguishing chromatin peak signals between wild type and mutant, we sought to quantify more subtle differences in chromatin accessibility, namely in putative binding of Transcription Factors (TFs). Using the TOBIAS framework [29], we computed base-pair footprint scores (relative depletion in accessible chromatin) for consensus peaks in the wildtype and mutant samples. Differences between these footprint scores at TF-motif occurrences within consensus peaks were then used to compute differential binding scores for each mouse motif available in the Jaspar database [30].
** Updated text **
Because we found no strong distinguishing chromatin peak signals between wild type and mutant, we investigated whether our ATAC-seq datasets can provide evidence of more subtle effects and in particular differences in TF binding. To do this we use genomic-footprinting analyses, which exploit the fact that the presence of bound TFs can protect DNA against transposase cleavage, resulting in relative decreases of accessibility signal within accessible regulatory elements. Using the TOBIAS framework [29], we computed base-pair footprint scores (relative depletion in accessible chromatin) for consensus peaks in the wildtype and mutant samples. Differences between these footprint scores at TF-motif occurrences within consensus peaks were then used to compute differential binding scores for each mouse motif available in the Jaspar database [30].
We also added an additional panel showing motif footprints for Vax1 into supplemental fig3, with amended legend.
Supplemental Figure S3. Motif footprints in wildtype and mutant samples.
(A) Aggregate Tn5-bias corrected ATAC-seq signal around detected Sox2 motif occurrences in consensus peak regions for wildype (green line) and mutant (red line) samples. (B) Same as for A for Vax 1 motif occurrences. Vertical dashed lines indicate edges of the Sox2 (A) and Vax1 (B) motifs.